# *Ad Hoc* Bayesian Program Learning

**Eli Sennesh**
Khoury College of Computer Sciences, Department of Psychology
Northeastern University
Boston, MA 02115
esennesh@ccis.neu.edu

## Abstract

Bayesian program learning provides a general approach to human-level concept learning in artificial intelligence. However, most priors over powerful programming languages make searching for a high-scoring program intractable, and therefore cognitively unrealistic. We hypothesize that an efficient learner searches programs which efficiently generate a likelihood by running to completion, and model this hypothesis with an *ad-hoc proposal* for programs. Our proposal works backwards from observations to find programs which quickly generate similar results.

## 1 Introduction

Bayesian models of cognition are a foundational paradigm for "reverse engineering the mind" as a route towards more human-like artificial intelligence [Tenenbaum et al., 2011, Marcus and Davis, 2013]. The theory of *ad hoc* cognition claims that humans spontaneously invent new concepts or theories as situations demand, without necessarily conforming to any *a priori* structure [Casasanto and Lupyan, 2015]. This theory has no standing analysis in Bayesian terms, but it does explain the flexibility of human cognition: every concept and theory is constructed and adjusted anew in each usage. The most radical Bayesian approaches to the mind have been based in program induction [Ritchie et al., 2018, Saad et al., 2019], explaining the human capacity for conceptual combination and compositional reasoning by positing that all concepts have the form of probabilistic programs, including abstract number concepts [Piantadosi et al., 2012] and grounded sensory concepts [Overlan et al., 2017]. We typically take the Church-Turing thesis as grounds to treat programs as a universal hypothesis class, and impose a prior over them based on either a probabilistic context-free grammar or the program's minimum description length [Vitányi and Li, 2000].

Models of program induction have mostly relied on domain-specific priors that encoded informed priors over task-relevant programs, as well as restrictions on general recursion [Ellis et al., 2018]. These models assume desirable program properties are provable *a priori* , an approach limited by Rice's Theorem. In contrast, humans can infer generally recursive concepts [Lake and Piantadosi, 2019], and may be employing resource-rational reasoning [Lieder and Griffiths, 2019].

Here we consider Bayesian program induction over a simple language and its evaluation semantics.

### 1.1 Contributions

We introduce a new proposal distribution for typed programs based on a novel inductive bias: resource-rational programs have short traces. This proposal allows us to quickly locate programs generating the observed data with reasonable marginal probability. We use a grammatical prior with our novel proposal to perform inference by importance sampling.

Preprint. Under review.

$$c ::= \mathbb{N} \mid \mathbb{R} \mid \mathbb{S} \mid \mathbf{true} \mid \mathbf{false} \qquad (1)$$

$$\mathtt{operator} ::= + \mid - \mid * \mid / \qquad (2)$$

$$e ::= v \mid (e_1 \, e_2) \mid \lambda v : \tau.e \mid \mathbf{flip} \, e \mid c \mid (\mathtt{operator} \, e_1 \, e_2) \mid \qquad (3)$$

$$\mathbf{if} \, e_1 \, \mathbf{then} \, e_2 \, \mathbf{else} \, e_3 \qquad (4)$$

$$\tau ::= \mathbb{N} \mid \mathbb{R} \mid \mathbb{S} \mid \{\mathbf{true}, \mathbf{false}\} \mid \tau_1 \to \tau_2 \qquad (5)$$

Table 1: Grammar of types $\tau$ and expressions $e$ in our simply-typed lambda calculus

## 2 An *Ad Hoc* Proposal for Typed Programs

In order to propose programs capable of quickly explaining observations, we assume the following:

- Any program we sample as a hypothesis should be guaranteed to run and score the data without throwing an exception or otherwise diverging. We guarantee this via type-safety.
- Evaluating a program produces an evaluator trace which assigns the data positive probability density.
- Traces should be relatively short to minimize necessary computation.

We employ a grammatical prior over program expressions, conditioned upon the static type of the desired expression. Given a type from which to sample a term, we choose uniformly at random from the set of expression constructors inhabiting that type (corresponding to grammar rules given in Table 1), then recursively fill in any necessary subexpressions. Constants are drawn from the uniform discrete distribution (Booleans), a recursive uniform-discrete sampler over the English alphabet (strings), the geometric distribution (integers), and the standard normal distribution (double-precision floating point numbers). Only $\lambda$-abstractions and bound variables can have functional type.

### 2.1 Small-step operational semantics

Computer scientists usually specify evaluation functions as recursive reductions from expressions to values, lumping many evaluation steps into one reduction. In order to neatly invert our evaluator, we specify it in terms of *small-step* semantics: a set of rules which can be (nondeterministically) applied to reduce expressions to expressions, eventually reaching a normal form.

### 2.2 Preservation and progress: ensuring a trace exists

To ensure that we can exactly invert the reduction rules to obtain expansion rules, every reduction rule must either reduce or syntactically reject every possible expression: we must never sample an expression that becomes "stuck" before reaching a normal form.

This property is formalized in programming languages as type *soundness*, consisting of preservation and progress properties. Preservation is the guarantee that all reduction rules syntactically applicable to an expression preserve the type of the expression. Progress tells us that well-typed expressions are not stuck: they are either normal forms, or at least one reduction rule can be applied to move towards a normal form of the appropriate type. Type soundness can be proved for the simply-typed lambda calculus in a standard fashion [Pierce et al., 2018].

### 2.3 *Ad hoc* inference as inference over traces

Given the small-step semantics for a typed programming language, one can trivially select any well-typed program and run it forward, obtaining its trace (the sequence of semantic rule applications leading to normalization and a final value). Note that while normal forms (values) reduce only to themselves, ensuring termination at the end of a reduction sequence, the same normal form can be reached starting from arbitrarily complex expressions.

We handle this inverse inference problem probabilistically: we invert the reduction rules to obtain expansion rules, and at each expansion step, randomly select an expansion rule whose input pattern

| Forward reduction rule for coin-flip operator |
|---|
| $p \in (0,1) \qquad val \sim \text{Bernoulli}(p)$ |
| $\text{flip}\,p \mapsto val$ |
| **Inverse expansion rule for coin-flip operator** |
| $v \in \{\textbf{true}, \textbf{false}\} \qquad p \sim \text{Uniform}(0,1)$ |
| $v \rightsquigarrow \textbf{flip}\, p$ |

Table 2: Example reduction and expansion rules for a **flip** operator

```
if (flip 3.059908452863891e−2) then
    True
else
    False
```
```
if False then
    flip 0.5897988150745864
else
    if True then
        flip 0.8754357227441879
    else
        flip 0.8838256505797507
```
```
flip 0.8928934208665623
```

Table 3: Ad-hoc programs inferred based on flips of a biased coin

(the outputs of the corresponding reduction rule) syntactically match the current expression. Since these expansion rules are the exact inverses of the reduction rules, our proposal thus denotes a distribution over expressions yielding a fixed value.

In a probabilistic lambda calculus, as we apply here, the resulting expressions are guaranteed to include the desired final value in their support set, evaluating to it with a nonzero likelihood.

## 3   Experiments

We performed two experiments, each consisting of presenting several observations and using self-normalized importance sampling to obtain ten sample programs inferred by small-step expansion.

In our first experiment, our model learned programs describing a biased coin, based on observations of $[\textbf{true}, \textbf{true}, \textbf{true}, \textbf{false}]$. In our second experiment, we used the natural numbers $[0, 1, 2, 3, 4]$ as observations, hoping to sample programs which would narrowly explain our chosen range of integers.

## 4   Results

In our experiment where we presented the results of a biased coin-flip as observations, our inference process yielded programs such as those found in Table 3. These programs appear to do the job we were trying to model: they flip a coin with a strong, but not total, bias towards heads.

In our second experiment, we inferred ad-hoc programs based on observations of the first five natural numbers, the sequence 0...4. The programs inferred to the sequence can be found in Table 4.

Many of the programs sampled for each observed problem were quite large, and involved passing short first-class functions as arguments. We believe this indicates that our prior currently does not sufficiently penalize the proposed programs for their program size in the importance resampling process. We also found that many of the sampled program fragments encoded information they did not use to compute their results, as in the following example.

```
((λi. 0.37875477753557496) "t")
```

```
if ( if False then True else False ) then
    1/((2*2)*(λj.1  λu.  True))
else
    ( if True then 9 / 3 else 3)
```

| $0 - 0$ |
|---|
| $2$ / $(0 + (/ 1 1))$ |
| $2 - 2$ |

Table 4: Programs inferred from the first five natural numbers

## 5   Discussion

Well-typed programs do not "go wrong", in precisely the sense of preservation and progress theorems [Milner, 1978]. Here we have exploited this guarantee to search the space of programs in a typed language *backwards*, from an observed output, to an expression which quickly generates that output with relatively high probability. We have made our source code available[1] for reproducibility.

However, these results show that our proposal currently does not have any helpful bias towards expanding only "interesting" programs, in which every subexpression contributes significantly to the output. We will outline several paths towards this goal, in addition to general improvements to our technique. A mechanism such as variable-less "higher-order" Bayesian networks [Overlan, 2019] would provide more efficient re-use of typed subexpressions than our lambda calculus' named variables. If we assume our observation comes from a primitive type and similarly equip each primitive type with a sampler covering all its values as support, we can also select the appropriate typed random sampler as a final likelihood and expand by inverse evaluation from there, turning our proposal into a prior. Since this prior would expend its probability mass on choosing and running small-step expansion rules, it would apply the Bayesian Occam's Razor to find programs which run in a short sequence of reduction steps. This may provide a novel probabilistic model of bounded rationality. Similarly, any prior or proposal over programs could be trained via wake-sleep methods, despite its use of discrete random choices [Anh et al., 2019]; vector types and neural networks could also be added to the program language. Finally, we could reframe the program-induction problem as one of program approximation [Kerinec et al., 2018, Dal Lago and Leventis, 2019].

## 6   Acknowledgments

The authors would like to thank Adam Scibior for his help with the details of monad-bayes, and Luke Hewitt for his guidance on the work described here.

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
