# OpenReview forum: "Ad-Hoc Bayesian Program Learning"
_NeurIPS.cc/2019/Workshop/Program_Transformations — Program Transformations @NeurIPS2019 Poster_

### Official Review · AnonReviewer2 · 2019-09-28
**Writing could be reworked for a more general audience; topic appears interesting and within scope**

**Confidence:** 1
**Rating:** 6

**Review:**

This paper deals with Bayesian program induction for a simple language. The introduction motivates the paper with Bayesian models of human cognition, but does not return to this topic, and the connection between this motivation and the rest of the work could use further explanation. The submission is half a page under the limit, and this space would be well-utilized by adding additional context and explanation to make it understandable to a more general audience. This paper is far from my area of specialty, and I can’t give a good evaluation of its technical contribution, but it seems to be interesting and within the scope of the workshop.

---

### Official Review · AnonReviewer1 · 2019-09-29
**Interesting ideas, missing closely related work**

**Confidence:** 3
**Rating:** 6

**Review:**

This paper proposes an interesting idea, but is missing discussions of two things:
1) Schmidhuber's and other's work on the speed prior, which exactly treats the idea of priors on programs with short traces.  There are results on how this compares to the Solomonoff prior.
2) More discussion of how approximate inference would work in this model.  Clearly just sampling backwards traces will not scale, which is the problem that inference in probabilistic programming has been trying to address for 10 years.  There are some citations at the end, but this is the central issue of this approach.  It's also not clear that the type system is making anything easier than existing PP systems.

---

### Decision · Program_Chairs · 2019-10-01

**Decision:**

Accept (Poster)

**Comment:**

A good contribution but the reviewers noted some problems with the exposition of the work.